# Giving It a Shot with a Different Approach: Prosocial Strategies Moderate the Joint Effects of Agentic and Communal Goals on Bullying

**DOI:** 10.3390/bs14070583

**Published:** 2024-07-10

**Authors:** Yangan Wang, Qingqin Zhang, Zixiao Dong, Xiangkui Zhang

**Affiliations:** School of Psychology, Northeast Normal University, Changchun 130024, China; wangyangan@nenu.edu.cn (Y.W.); zhangqingqin139@nenu.edu.cn (Q.Z.); zxdong@nenu.edu.cn (Z.D.)

**Keywords:** bullying, social goals, agency, communion, prosocial strategies, resource control strategies, response surface analysis, adolescent

## Abstract

This study uses polynomial regression and response surface analyses to investigate the joint effects of agentic and communal goals on bullying and the moderating role of prosocial strategies. The sample included 917 adolescents (M_age_ = 13.54, *SD* = 1.02) from rural, suburban, and urban areas in China. The findings revealed that higher agentic and lower communal goals were associated with a linear rise in bullying. Surprisingly, when both social goals were higher simultaneously, bullying followed an inverted U-shaped pattern. Furthermore, prosocial strategies moderated the joint effects of the two social goals. Adolescents who are more likely to use prosocial strategies do not show significant changes in bullying when both goals are at a higher level. In contrast, those who are less likely to do so show a linear rise in bullying, regardless of changes in social goals. This study improves our understanding and intervention of bullying behavior, emphasizing a non-pathological perspective.

## 1. Introduction

Bullying in schools is a significant global issue [1]. It peaks in early adolescence and then decreases in late adolescence [2]. Research based on social cognition and evolutionary psychology emphasizes the adaptiveness of bullying as a goal-directed behavior carried out in the context of power imbalances, serving goals that transcend others. At the same time, many researchers increasingly highlight its adaptive and strategic facets, advocating for a non-pathological view [3,4,5,6,7,8]. In a thought-provoking study, Hensums et al. [4] incorporated two social goals, agency, and communion, into their model and examined their relationship to bullying. They observed a positive correlation between agentic goals and bullying behavior, whereas no significant relationship was identified between communal goals and bullying. The findings suggest that communal goals may only predict bullying behavior negatively when agentic goals are low or when communal goals exceed a specific threshold. Therefore, studying the joint effects of both types of social goals on bullying is essential. However, their study was limited by the methodology, only examining the independent effects of these goals; they could not determine their joint effects. What would be the effect of simultaneously heightening or lowering agentic and communal goals on bullying? To address such questions, a response surface analysis (RSA), based on polynomial regression, can effectively examine the complex relationships among three variables [9].

From an evolutionary psychology perspective, bullying is seen as adaptive, as it can help individuals gain resources and advantages in competitive environments. This does not mean we should tolerate bullying, but rather that we should understand the underlying psychological processes to develop effective interventions. The Meaningful Role Intervention (MRI) program suggests that many bullies seek social dominance and status. It also posits that these goals can be achieved through prosocial means, such as an information technology specialist or a homework monitor [10]. In light of the intervention program findings, we aim to explore how prosocial strategies can moderate the joint effects of two social goals. Using RSA, this article addresses these questions from social cognition and evolutionary psychology viewpoints.

### 1.1. The Relationship between Social Goals and Bullying

This study employs the social information processing theory (SIP) and a dichotomous framework of agency and communion to explore adolescents’ social goals. In SIP, a goal is a state of heightened arousal that points to a specific desired outcome [11]. Individuals approach situations with predefined goals and dynamically adjust them as the situation unfolds. Social goals reflect individuals’ expected outcomes in social interactions and relationships [4]. We use a dichotomous framework to understand adolescents’ social goals [12,13,14]—agentic goals for getting ahead and communal goals for getting along [15]. Agentic goals focus on gaining influence, status, and personal success, while communal goals center on establishing close, cooperative relationships and focusing on the needs and interests of others.

Agentic and communal goals reflect individuals’ motivations in social interactions and may influence behavior in several ways. First, some studies have found a positive relationship between agentic goals and bullying. Hensums et al. [4] found that agentic goals positively correlated with bullying, and Caravita and Cillessen [13] also reported consistent findings in the prior literature. Meta-analyses suggest that adolescents with higher rates of bullying behavior are more likely to endorse agentic goals [16]. Second, research on the relationship between communal goals and bullying presents a mixed picture. Studies by Hensums et al. [4] and Caravita and Cillessen [13] suggest no relationship between communal goals and bullying. However, other researchers have found a negative correlation between these variables [17]. Further research is needed to clarify the relationship between communal goals and bullying. In this paper, we hypothesize a positive correlation between agentic goals and bullying and a negative correlation between communal goals and bullying (Hypothesis 1).

### 1.2. The Joint Effects of Two Goals

Despite seeking power and dominance, a bully may have other goals. Individuals can simultaneously pursue high and low agentic goals and high or low communal goals without inherent opposition [18,19,20]. The existing literature debates the correlation between the two social goals [4,13,17,21,22,23,24,25]. When adolescents simultaneously hold congruent or incongruent social goals, the relationship between these two goals and their joint effect on bullying may change.

The previous section suggested that the two social goals show contrasting trends in their ability to predict bullying. Therefore, when both social goals heighten simultaneously, their effects on bullying may offset each other, leading to insignificant changes in bullying behavior. However, Hensums and colleagues point out that the relationship between agentic goals and bullying behavior may be stronger than that with communal goals. They found that agentic goals were positively correlated with bullying behavior, while communal goals did not show a significant correlation [4]. The findings demonstrate a direct association between agentic goals and bullying behavior, whereas communal goals do not directly predict reduced levels of bullying. Building on the existing literature, we hypothesize that the predictive effects of communal goals on bullying are nonlinear. This implies that communal goals may need a certain threshold to predict bullying behavior negatively [26]. Studying the quadratic effect of communal and agentic goals helps us grasp non-linear changes in their predictive role. Considering the asymmetrical relationship between the two types of social goals and bullying behavior, we formulate the following hypotheses based on the existing literature. Hypothesis 2a: As agentic goals ascend to higher levels and communal goals decline to lower levels, there is a significantly higher level of bullying. Hypothesis 2b: When agentic and communal goals reach higher levels simultaneously, bullying is heightened significantly rather than remaining unchanged.

### 1.3. The Moderating Role of Prosocial Strategies

Social cognition and evolutionary psychology offer a non-pathological explanation and intervention perspective for bullying [4]. This article responds to the call of Hensums et al. [4] by combining these two perspectives to examine the mechanisms of bullying. The MRI indicates that resources initially obtained through bullying behavior (e.g., status or influence in peer groups) can also be acquired more prosocially [10]. This perspective aligns with another theory based on evolutionary psychology, the Resource Control Theory (RCT).

The RCT examines how individuals use different strategies to gain material or social resources [27]. Children and adolescents use both prosocial and coercive strategies to obtain resources. Prosocial strategies include building cooperative relationships, making suggestions, offering help, and establishing social connections and networks. Coercive strategies involve coercion to get resources directly, such as forcefully taking toys, making threats, and exerting control over resources through coercive actions. Bullies might see bullying as a way to achieve their goals because they have learned that using coercive strategies has been influential in the past [10,28,29]. However, the MRI recognizes the strengths and goals of bullies and aims to guide them towards more positive activities to achieve their goals, which they had previously pursued through antisocial actions.

The RCT and MRI suggest that prosocial strategies could moderate the joint effect of agentic and communal goals on bullying. It is crucial to clarify that prosocial strategies are not aligned with altruism but rather with personal goal attainment [30]. These strategies can indeed benefit others, but they differ from prosocial behavior aimed at benefiting others [31]. Prosocial strategies are tools for gaining resources, achieving social goals, and enhancing status. Some studies have found that prosocial strategies correlate positively with agentic goals, weakly with bullying, and are unrelated to communal goals [17,32]. These results indicate that prosocial strategies are more closely related to agentic goals and serve in personal goal attainment. Since coercive strategies are closely related to agentic goals [17], adolescents with high prosocial strategies and agentic goals may be dual-strategy controllers who are highly socially skilled. Hence, we propose that individuals with high levels of prosocial strategies and agentic goals but low levels of communal goals may employ various means to access what they want. These means may include ingratiation, exclusion, or relational aggression [32]. In light of the existing literature, we put forth Hypothesis 3a: As agentic goals reach higher levels and communal goals reach lower levels for adolescents likely to employ prosocial strategies, bullying will significantly heighten. In the above hypothesis, prosocial strategies do not appear to prevent bullying behavior in adolescents with high agentic and low communal goals. However, the moderating effect of prosocial strategies may change when both social goals heighten simultaneously. Specifically, although agentic goals may initially overshadow communal goals [22,33], the effects of communal goals may become more significant at this stage. This change occurs because adolescents who use prosocial strategies highly value maintaining interpersonal relationships. In these conditions, the effects of the two social goals on bullying may cancel out each other. As a result, their bullying behavior may not change significantly. Therefore, we propose Hypothesis 3b: When adolescents with high levels of agentic and communal goals utilize more prosocial strategies, no significant change in their bullying behavior will occur.

### 1.4. The Present Study

This research utilizes social cognition theories and evolutionary psychology to investigate the relationships among social goals, bullying, and prosocial strategies. It focuses on agentic goals, emphasizing influence and dominance, and communal goals, which aim to build positive relationships. The study explores how these social goals are associated with bullying behavior and examines the moderating role of prosocial strategies. Using polynomial regression and RSA, the researchers test these associations. This study aims to improve our understanding of the psychological processes underlying bullying and offer insights into effective interventions for reducing bullying in schools.

## 2. Methods

### 2.1. Participants and Procedure

The present study included 917 valid participants (M_age_ = 13.54, *SD* = 1.02) who passed the attention check. We excluded 119 participants from the original dataset due to failing the attention check or selecting identical responses for all items. The participants were from northern and southern China, distributed across 29 classes in three distinct rural, suburban, and urban schools. The sample included 169 grade 5 students (46.2% female), 201 grade 6 students (60.7% female), 288 grade 7 students (53.3% female), and 259 grade 8 students (57.5% female).

The entire data collection procedure was conducted over two days. On the first day, the study details were presented to the principals of the three schools via telephone, and their permission was secured. Afterward, the principals arranged for the teachers to present the study to the students. Subsequently, the students were instructed to inform their parents about the situation to ensure parental awareness. Parents who chose not to have their children participate were advised to promptly contact the school to express their concerns. If there were no objections, the students decided to participate in the study independently. This method has been used in prior studies [34]. Ten parents withdrew their consent for their children to participate in the study. Before starting the survey on the second day, the research assistants gave the teachers clear instructions on how to help the students complete the questionnaire. The students were then told that their answers would be kept confidential and that they could opt out of the study at any time. It took about 30 min to finish the questionnaire. After completing the survey, all the students received a gift package, including a black pen, a transparent file folder, and a cartoon-themed notepad. Those who did not participate in the study received the same gifts. This study was approved by the Ethics Committee of the author’s institution.

### 2.2. Measures

#### 2.2.1. Social Goals

This study adapted the Social Goals Questionnaire (SGQ), initially developed by Jarvinen and Nicholls [35] and subsequently revised in its Chinese version by Zou and Lin [36]. The original scale comprised six dimensions: dominance, intimacy, nurturance, leadership, popularity, and avoidance. In line with the theoretical literature, this study categorized dominance and leadership as agentic goals (10 items, e.g., “*I like it when I make them do what I want*”, “*I like it when they say I’m the boss*”; α = 0.78), and intimacy and nurturance as communal goals (11 items, e.g., “*I like it when we know each other’s private feelings*”, “*I like it when I go out of my way to help them*”; α = 0.85), thereby resulting in a two-factor structure [14]. The scale employed a 5-point Likert scale (1 = completely not true, 5 = completely true), with higher total scores indicating a stronger tendency for individuals to hold those social goals. A confirmatory factor analysis (CFA) revealed a good fit of the model: χ^2^/*df* = 2.670, *SRMR* = 0.048, *RMSEA* = 0.043, *GFI* = 0.954, *NFI* = 0.921, *CFI* = 0.949 (α = 0.78).

The popularity subdimension was excluded for two reasons. Firstly, Chinese adolescents do not prioritize popularity goals in the development of their social status [37], and its inclusion in the agentic goals factor significantly deteriorated the model fit (χ^2^/*df* = 5.563, *SRMR* = 0.105, *RMSEA* = 0.071, *GFI* = 0.852, *NFI* = 0.785, *CFI* = 0.816). Secondly, there is a certain degree of ambiguity regarding whether popularity belongs to agentic or communal goals within the dichotomous framework [38]. Additionally, the avoidance subdimension was omitted because it does not align with this study’s definition of social goals.

#### 2.2.2. Bullying

The revised Chinese version of the Olweus Bullying Questionnaire was employed to measure physical, verbal, and relational bullying [39]. The participants were queried about the frequency of engaging in various categories of bullying behaviors over the past few months. The scale consists of 6 items, using a 5-point rating scale (0 = *never*, 4 = *several times a week*), with higher scores indicating a greater actual incidence of bullying (e.g., “*Like kicking, hitting, shoving or running into other students*”). A CFA revealed a good model fit: χ^2^/*df* = 4.127, *SRMR* = 0.022, *RMSEA* = 0.058, *GFI* = 0.991, *NFI* = 0.980, *CFI* = 0.985 (α = 0.77).

#### 2.2.3. Prosocial Strategies

An adapted scale of prosocial strategies has been developed based on existing research, designed for participants to self-assess the extent to which they employ such strategies [40,41,42,43]. The scale comprises 8 items (e.g., “*I cooperate with others to work together for what we want*”). The participants responded to these items using a 5-point Likert scale (1 = *completely not true*, 5 = *completely true*). A CFA revealed a good model fit: χ^2^/*df* = 3.431, *SRMR* = 0.029, *RMSEA* = 0.052, *GFI* = 0.982, *NFI* = 0.963, *CFI* = 0.974 (α = 0.80).

#### 2.2.4. Covariate

In the subsequent analysis, age and gender were included as covariates because they were beyond the scope of this study but had the potential to influence the results. It is worth noting that bullying behavior is often associated with coercive strategies and can be seen as a specific form of such strategies [44]. People who display bullying behavior are usually either coercive or use both coercive and prosocial strategies [5]. To properly differentiate between bullying behavior and coercive strategies, it is vital to consider coercive strategies as a covariate. Also, due to the RSA software’s technical limitations (Version 4.0.3), it is impossible to include two moderating variables simultaneously. The coercive strategies scale was constructed using the same procedure as the prosocial strategies scale (6 items total; e.g., “*I try to force others to follow my plans*”). A CFA demonstrated a good model fit: χ^2^/*df* = 4.370, *SRMR* = 0.018, *RMSEA* = 0.061, *GFI* = 0.992, *NFI* = 0.987, *CFI* = 0.990 (α = 0.82).

#### 2.2.5. Attention Check

We developed an instructional attention check item (“Respond with ‘completely not true’ for this item”) to exclude careless and insufficient effort responses [45].

#### 2.2.6. Data Analyses

The variable data were normalized to standardize the scales and reduce the risk of multicollinearity. We used a macro program for SPSS to perform the RSA, and response surface curve graphs were created using Excel (Version 2019 MSO 16.0.10408.20002) [46]. The analysis process was divided into two main components. First, preliminary analyses, including descriptive statistics and correlation analysis, were conducted to provide evidence for subsequent analysis. Subsequently, the RSA was employed to investigate the interactions between variables further, specifically by establishing a polynomial regression model with bullying behaviors (*Z*) as the dependent variable. The model was as follows:*Z* = *b*_0_ + *b*_1_*X* + *b*_2_*Y* + *b*_3_*X*^2^ + *b*_4_*XY* + *b*_5_*Y*^2^ + *b*_6_*W* + *b*_7_*WX* + *b*_8_*WY* + *b*_9_*WX*^2^ + *b*_10_*WXY* + *b*_11_*WY*^2^ + *e*

Here, *X* represents agentic goals, *Y* represents communal goals, *W* represents prosocial strategies, and *b*_0_ is the constant term. Coefficients *b*_1_ to *b*_5_ represent the non-standardized regression coefficients, respectively, representing the linear effects of agentic goals (*b*_1_), the linear effects of communal goals (*b*_2_), the quadratic effects of agentic goals (*b*_3_), the interaction effects of agentic and communal goals (*b*_4_), and the quadratic effects of communal goals (*b*_5_). Furthermore, coefficients *b*_6_ to *b*_12_ indicate that the effects of *b*_1_ to *b*_5_ are moderated by *W*. If the model’s determination coefficient Δ*R*^2^ significantly increases after adding these moderating terms, this suggests that prosocial strategies have a significant moderating effect on bullying behaviors [9,47,48].

## 3. Results

### 3.1. Common-Method Bias

Harman’s single-factor test was employed to assess the potential for common-method bias. The analysis identified 10 common factors with eigenvalues greater than 1. The first common factor explained 21.09% of the variance, falling below the critical threshold of 40%. Consequently, this study was found to have no significant common-method bias.

### 3.2. Preliminary Analyses

Table 1 displays the means, standard deviations, and correlation coefficients among the study variables before standardizing the variables. The results indicated that agentic goals were positively correlated with bullying, whereas communal goals and prosocial strategies were unrelated to agentic goals. In contrast, communal goals were negatively correlated with bullying but positively correlated with prosocial strategies. Conversely, prosocial strategies were negatively correlated with bullying.

### 3.3. Main Analyses

To assess the response rate of the sample and determine if it met the necessary standards for RSA, a criterion of 0.5 standard deviations was used to analyze differences in the levels. The analysis showed that among the sample, 33.48% had higher scores for agentic goals than communal goals, 40.79% had lower scores for agentic goals than communal goals, and 25.74% had equal scores. This distribution suggests that the sample was relatively uniform and appropriate for RSA [49,50].

Table 2 presents the results of a polynomial regression analysis with bullying as the dependent variable. Model 2 showed that agentic goals had a significant and positive correlation with bullying (*b* = 0.11, *t* = 2.64, *p* < 0.01), while communal goals had a significant and negative correlation with bullying (*b* = −0.13, *t* = −4.16, *p* < 0.001). Similar patterns were observed in Models 3 through 5, supporting H1.

### 3.4. The Joint Effects of Social Goals on Bullying

In the next step, we analyzed the slopes, curvatures, and response surface curves based on the findings of Models 3 to 5 (refer to Table 1) to test the remaining hypotheses. According to Edwards and Parry (1993), the interpretation of response surface curves is based on four curvature test values: *a*_1_ (=*b*_1_ + *b*_2_), *a*_2_ (=*b*_3_ + *b*_4_ + *b*_5_), *a*_3_ (=*b*_1_ − *b*_2_), *a*_4_ (=*b*_3_ − *b*_4_ + *b*_5_). *a*_1_ represents the relationship between the two independent variables and the dependent variable when they change along the line of congruence. *a*_2_ indicates whether this relationship is linear or curvilinear. *a*_3_ represents the relationship between the independent and dependent variables when they change along the incongruence line. *a*_4_ indicates whether this relationship is linear or curvilinear.

Table 3 presents the results of all the slope and curvature analyses. When agentic and communal goals changed along the line of congruence, the slope was not statistically significant (*a*_1_ = 0.03, *t* = 0.50, *p* > 0.05), but the curvature was significant (*a*_2_ = −0.10, *t* = −2.47, *p* < 0.05). This study found that as both agentic and communal goals reached higher levels simultaneously, bullying displayed a non-linear, inverted U-shaped pattern. Specifically, when both social goals were initially low, bullying heightened as they simultaneously reached higher levels, peaking at the average level. However, as agentic and communal goals exceeded the average level and continued to rise, bullying began to decline (see Figure 1a,b). These findings partially supported hypothesis H2b. When the two independent variables were incongruent, the slope was significant (*a*_3_ = 0.28, *t* = 4.54, *p* < 0.001) and the curvature was not significant (*a*_4_ = 0.09, *t* = 1.94, *p* > 0.05), indicating that higher agentic goals coupled with lower communal goals correlated with a linear escalation in bullying, supporting H2a (see Figure 1a,c).

### 3.5. The Moderating Role of Prosocial Strategies

The results of the polynomial regression analysis showed that including the five moderating terms improved the model’s explanatory power by 2% (Δ*R*^2^ = 0.02, *F* = 15.64, *p* < 0.01), confirming the validity of the moderated RSA model. As shown in Table 4, prosocial strategies (PS) were categorized as either low or high based on one standard deviation.

When PS is equal to −1, the following values are obtained: *a*_1_ = 0.21 (*t* = 2.08, *p* < 0.05), *a*_2_ = −0.06 (*t* = −0.97, *p* > 0.05), *a*_3_ = 0.32 (*t* = 3.33, *p* < 0.001), *a*_4_ = 0.01 (*t* = 0.14, *p* > 0.05). The results indicated that adolescents who had a low likelihood of using PS showed a high level of bullying when their agentic and communal goals were elevated simultaneously. Concurrently, higher agentic goals and lower communal goals were associated with a heightened level of bullying.

When PS = 1, *a*_1_ = 0.02 (*t* = 0.31, *p* > 0.05), *a*_2_ = 0.03 (*t* = 0.40, *p* > 0.05), *a*_3_ = 0.11 (*t* = 1.23, *p* > 0.05), *a*_4_ = 0.25 (*t* = 3.11, *p* < 0.01). The preceding results indicated that for individuals who were more likely to use PS, there was no significant change in bullying as both agentic and communal goals reached higher levels simultaneously (see Figure 2a,b). This finding supported hypothesis H3b. Conversely, for this group of adolescents, bullying displayed a U-shaped pattern when two types of social goals were in a state of incongruence. In particular, when agentic goals were lower than communal goals, bullying lessened as agentic goals became higher and communal goals became lower, reaching a minimum at average levels of both goals. Conversely, when agentic goals were higher than communal goals, bullying heightened as agentic goals continued to rise and communal goals continued to fall (see Figure 2a,c). These findings provide partial support for H3a.

## 4. Discussion

Social goals represent individuals’ preferences for social outcomes, interpersonal needs, and values. This suggests that individuals prioritize goals differently, which leads to varying associated behaviors [36,51]. The primary objective of this study is to examine the joint effects of agentic and communal goals on bullying. This research combines social cognition and evolutionary psychology [4,6,10,11] to clarify the joint effects of these factors on bullying. Additionally, this study aims to explain the moderating role of PS on the joint effects. This study is the first to use RSA to examine linear and nonlinear relationships between social goals and bullying. After performing RSA on the scores of various variables for 917 adolescents, it was observed that there are indeed joint effects of these two social goals on bullying and that PS moderates these effects. Some results supported prior hypotheses, while others revealed unexpected findings.

### 4.1. The Relationship between Social Goals and Bullying

This study’s results supported Hypothesis 1, which suggests a positive correlation between agentic goals and bullying and a negative correlation between communal goals and such behavior. The findings align with those of Pronk et al. [17] but are in contrast to the results of Hensums et al. [4] and Caravita and Cillessen [13]. Notably, Hensums et al. [4] initially expected to find a negative relationship between communal goals and bullying. Acknowledging that the correlations presented here pertain to the linear associations between social goals and bullying is crucial. The subsequent sections of this paper will examine the interaction between agentic and communal goals and their relationship to both linear and nonlinear dynamics in bullying.

### 4.2. The Joint Effects of Two Goals on Bullying

This study found that higher agentic and lower communal goals were associated with a linear rise in bullying. This supports Hypothesis 2a. Additionally, Hypothesis 2b predicted that even when both agentic and communal goals are high, there would be more bullying instead of remaining unchanged. The initial hypothesis suggested a linear correlation, but further analysis revealed an inverted U-shaped pattern in bullying levels when both social goals were heightened simultaneously. Specifically, when both social goals were low, bullying was proportional to the elevation of agentic and communal goals. When agentic and communal goals reached average values, the peak incidence of bullying was observed. However, as both social goals continued to rise, the incidence of bullying began to decline. 

Adolescents who prioritize both agentic and communal goals exhibit less bullying behavior compared to those who prioritize agentic goals over communal goals. These individuals show no significant difference in bullying levels compared to peers who are indifferent to both types of goals. Finding a balance between agentic and communal goals is beneficial for adolescents. Those who pursue both goals can better adapt and be flexible in social situations, adjusting their behavior based on the circumstances [18,52]. For example, they can demonstrate agency to gain status or influence and show commission to build friendships or trust. This approach indicates that seeking various social resources may decrease their need to resort to bullying or aggression [30]. Therefore, adolescents pursuing both goals strive for status, power, or leadership while ensuring appropriate actions. Conversely, those scoring lower on both two goals may withdraw socially (e.g., characterized by submissiveness and separation in the interpersonal circumplex model) [14,53]. Previous studies have shown either a negative or no correlation between this combination of goals and aggression [14]. Even though aggression and bullying are different [3], understanding this relationship can help us explain why there is no significant difference in bullying behavior between individuals who actively pursue both goals and those who show no interest in either goal. Based on these findings, it is suggested that teenagers who pursue both agentic and communal goals and those who are indifferent to both goals may experience different outcomes in terms of their social status. Future research could explore this hypothesis further. Hensums et al. [4] proposed two predictions in their study. The first prediction suggested that communal goals could act as a buffer against bullying only when agentic goals were low. The second prediction was that the relationship between agentic goals and bullying would weaken when communal goals were high. This study’s results confirmed both of these predictions. In conclusion, this study’s findings showcase the advantages of using RSA and shed light on the intricate associations between agentic and communal goals and bullying, encompassing linear and nonlinear relationships [9,46]. Furthermore, the findings support a non-pathological view of bullying by indicating that individuals who engage in bullying perceive this behavior as a way to acquire resources such as status and dominance. This aligns with previous research [4,5,6].

### 4.3. The Moderating Role of Prosocial Strategies

As a variable of interest in this study, PS plays a role as a moderating variable. Firstly, the results of the moderated RSA partially supported Hypothesis 3a. The term “partially” is employed here because the hypothesis predicted a linear relationship, but the final results revealed a nonlinear trend. For adolescents less inclined to use prosocial strategies, bullying reached a higher level when their agentic goals exceeded their communal goals. This linear upward trend in bullying persisted, even when both types of social goals were balanced or when agentic goals were more dominant than communal goals. Conversely, adolescents who were more prone to employ PS showed a U-shaped trend in bullying under comparable circumstances. When these adolescents’ agentic goals are lower than their communal goals, bullying behavior declines as they strive for higher agentic goals and lower communal goals. The lowest point is reached when both types of social goals are moderately balanced. However, as agentic goals gradually overtook communal goals, bullying began to escalate once more. These results are similar to an O. Henry ending—both surprising and plausible. Ojanen et al. [14] suggested that an individual’s low agentic and high communal goals may indicate a quest for peer acceptance. This is achieved by aligning with others’ perspectives and meeting their expectations, which can enhance the tendency towards submissive and conformist behavior [54]. This indicates that individuals with a tendency for such conduct, who are more likely to use PS to obtain resources (such as collaborating with classmates and avoiding disputes), may also be more likely to submit to the bullying of their friends or peers or to serve as an assistant to the bully. Indeed, once the role of the assistant was included as a covariate, the moderated RSA model ceased to be statistically significant (see Appendix A).

This study confirmed Hypothesis 3b, showing no significant change in bullying when adolescents more likely to use PS had both agentic and communal goals elevated simultaneously. Conversely, when agentic and communal goals reach higher levels in lower PS, bullying behavior rises linearly. In conclusion, these findings align with the MRI [10]. The MRI suggests that guiding adolescents who bully others towards engaging in more prosocial activities can help them achieve their agentic goals. The study also indicates that while PS is beneficial for adolescents with very high and low agentic goals, it may not be as helpful for those with excessively high communal goals (similar to unmitigated communion). Unmitigated communion refers to an excessive focus on the well-being of others and becoming overly involved in their lives [55]. This concept might help explain the current findings. According to self-determination theory, adolescents in this state may struggle to fulfill their autonomy and competence needs, and external factors can easily influence their behavior [56]. Unmitigated communion suggests an absence of self-distinction, increasing susceptibility to manipulation by others [57]. Therefore, adolescents inclined towards unmitigated communion may abandon their principles to seek peer acceptance, conform to peer norms, and potentially engage in bullying. It is crucial to consider how overly high communal goals relate to PS. Indeed, this inclination suggests a higher likelihood of defending behavior contingent upon one’s peer group. Future research could investigate this intriguing hypothesis.

### 4.4. Limitations

The current study has several limitations. Firstly, the data used in the analysis are cross-sectional, so we cannot infer causal relationships between variables. Collecting longitudinal data across different time points and using RSA in future research could help address this limitation. Secondly, the data for this study were obtained through self-report from the participants. While common-method bias did not affect the findings in this case, including data from multiple raters could make the conclusions more robust. Thirdly, communal goals and prosocial strategies had a moderate to high correlation coefficient in the correlation analysis (*r* = 0.66). However, another study also found a moderate to high correlation between communal goals and prosocial behavior in Chinese adolescents [58]. Although the purposes of prosocial strategies and prosocial behavior differ, their behavioral manifestations are similar. Fourth, this study used a one-dimensional measure of agentic and communal goals without distinguishing their facets. For example, we categorized dominance and leadership as agentic goals. However, the association between these two dimensions and bullying behavior might vary. Future research should investigate how various facets of agentic and communal goals affect bullying behaviors and their interactions.

### 4.5. Implications

Despite the limitations mentioned, this study provides valuable theoretical and practical insights. This study expands on previous research on bullying from social cognition and evolutionary psychology perspectives. It confirms the joint effects of agentic and communal goals on bullying and examines how prosocial strategies can moderate these effects. Additionally, we explored the linear and nonlinear relationships between the four variables. These findings contribute to a more comprehensive understanding of bullying behavior and its underlying mechanisms from a non-pathological standpoint, which can help develop practical interventions.

The results of this study also have implications for anti-bullying prevention and intervention programs. Firstly, considering the insights provided by the moderated RSA highlights the need to maintain a balanced development of agentic and communal goals, especially for adolescents more likely to use prosocial strategies to acquire resources [25,59]. According to the MRI and self-determination theory findings, encouraging adolescents’ involvement in diverse roles across different positions can effectively nurture and satisfy their basic psychological needs [28,60]. The three basic psychological needs (autonomy, competence, and social relatedness) are positively associated with agency and communion [61]. Otherwise, the positive effects of prosocial strategies would be diminished for adolescents with excessively low agentic and high communal goals.

Secondly, the significance of agentic goals is irrefutable during adolescence [12]. Adolescents are motivated to seek respect and status [62]. We agree with the viewpoint of Hensums et al. [4], which suggests that instead of reducing adolescents’ agentic goals, it is better to modify how they pursue them. The results of the MRI support this perspective. The purpose of the bully may be clear, but the means they use to achieve it are harmful to themselves and others and are not socially acceptable [63]. Therefore, it would be beneficial for bullies to be directed towards developing positive strategies to control resources, allowing them to pursue agentic goals in a way that benefits others rather than causing harm [10,28]. For example, giving bullies the responsibility of overseeing the class’s environmental initiatives, such as garbage sorting or acting as ecological monitors, allows them to gain status and recognition.

## 5. Conclusions

This study takes social cognition and evolutionary psychology perspectives to examine how agentic and communal goals are related to bullying and how prosocial strategies can moderate this relationship. Higher agentic and lower communal goals were associated with a linear rise in bullying. When both social goals were higher simultaneously, bullying followed an inverted U-shaped pattern. Adolescents who are more likely to use prosocial strategies do not show significant changes in bullying when both types of social goals are at a higher level. In contrast, those who are less likely to use prosocial strategies show a linear rise in bullying, regardless of changes in social goals.

## Figures and Tables

**Figure 1 behavsci-14-00583-f001:**
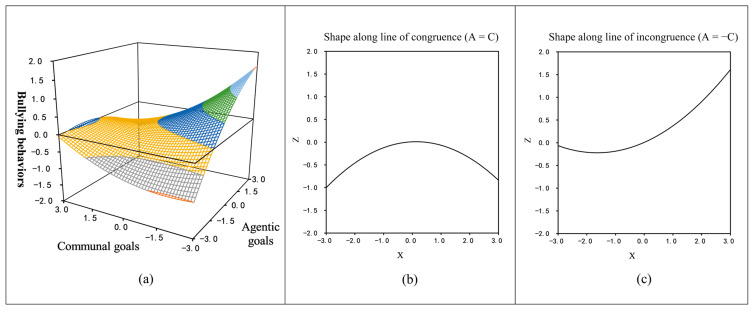
(**a**) The response surface curve graph; (**b**) a sectional diagram of the response surface’s shape along the line of congruence; (**c**) a sectional diagram of the response surface’s shape along the line of incongruence.

**Figure 2 behavsci-14-00583-f002:**
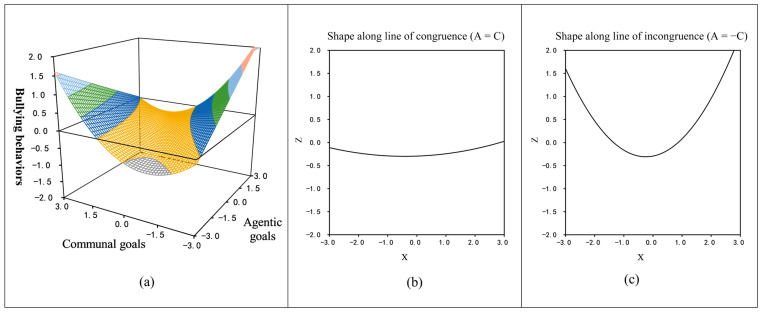
(**a**) The response surface curve graph; (**b**) a sectional diagram of the response surface’s shape along the line of congruence; (**c**) a sectional diagram of the response surface’s shape along the line of incongruence.

**Table 1 behavsci-14-00583-t001:** Descriptive statistics and correlation analysis (*N* = 917).

	*M*	*SD*	1.	2.	3.	4.	5.	6.	7.
1. Age	13.54	1.02	−						
2. Gender	–	–	−0.014	−					
3. Agentic goals	1.87	0.62	−0.169 ***	−0.075 *	−				
4. Communal goals	3.82	0.73	0.204 ***	0.149 ***	−0.005	−			
5. Bullying	1.15	0.34	−0.241 ***	−0.146 ***	0.247 ***	−0.204 ***	−		
6. Prosocial strategies	3.96	0.70	0.153 ***	0.054	−0.061	0.660 ***	−0.244 ***	−	
7. Coercive strategies	1.71	0.75	−0.165 ***	−0.075 *	0.680 ***	−0.132 ***	0.274 ***	−0.164 ***	−

* *p* < 0.05, *** *p* < 0.001, gender is coded as a dummy variable (1 = male, 2 = female).

**Table 2 behavsci-14-00583-t002:** Polynomial regression analysis (*N* = 917).

Variables	Bullying				
	Model 1	Model 2	Model 3	Model 4	Model 5
	*b* (*SE*)	*b* (*SE*)	*b* (*SE*)	*b* (*SE*)	*b* (*SE*)
Step 1					
Age	−0.20 (0.03) ***	−0.17 (0.03) ***	−0.17 (0.03) ***	−0.17 (0.03) ***	−0.16 (0.03) ***
Gender	−0.13 (0.03) ***	−0.11 (0.03) ***	−0.11 (0.03) ***	−0.12 (0.03) ***	−0.12 (0.03) ***
Coercive strategies	0.23 (0.03) ***	0.15 (0.04) ***	0.14 (0.04) ***	0.13 (0.04) **	0.13 (0.04) **
Step 2					
Agentic goals (A)		0.11 (0.04) **	0.15 (0.05) ***	0.15 (0.05) ***	0.17 (0.05) ***
Communal goals (C)		−0.13 (0.04) ***	−0.13 (0.04) ***	−0.03 (0.04)	−0.05 (0.05)
Step 3					
A^2^			−0.05 (0.02) *	−0.05 (0.02) *	−0.03 (0.02)
A × C			−0.09 (0.03) **	−0.09 (0.03) **	−0.07 (0.04)
C^2^			0.04 (0.02)	0.03 (0.02)	0.09 (0.04) *
Step 4					
Prosocial strategies (PS)				−0.16 (0.04) ***	−0.25 (0.05) ***
Step 5					
A × PS					−0.10 (0.05) *
C × PS					0.01 (0.04)
A^2^ × PS					0.04 (0.02)
A × C × PS					−0.04 (0.03)
C^2^ × PS					0.05 (0.02) **
*R* ^2^	0.13 ***	0.15 ***	0.17 ***	0.18 ***	0.20 ***
Δ*R*^2^	0.13 ***	0.02 ***	0.02 ***	0.01 ***	0.02 **
*F*	46.11	32.57	22.61	22.10	15.64
Δ*F*	46.11	10.79	5.24	15.22	3.48

* *p* < 0.05, ** *p* < 0.01, *** *p* < 0.001. The genders “male” and “female” are coded as 1 and 2, respectively.

**Table 3 behavsci-14-00583-t003:** Slope and curvature analysis (no moderator).

	Shape along the Line of Congruence: Agentic Goals = Communal Goals (A = C)		Shape along the Line of Incongruence: Agentic Goals = −Communal Goals (A = −C)	
Slope	*a* _1_	0.03	*a* _3_	0.28 ***
Curvature	*a* _2_	−0.10 *	*a* _4_	0.09

* *p* < 0.05, *** *p* < 0.001.

**Table 4 behavsci-14-00583-t004:** Slope and curvature analysis.

		Shape along the Line of Congruence: Agentic Goals = Communal Goals (A = C)		Shape along the Line of Incongruence: Agentic Goals = −Communal Goals (A = −C)	
PS = −1	Slope	*a* _1_	0.21 *	*a* _3_	0.32 ***
	Curvature	*a* _2_	−0.06	*a* _4_	0.01
PS = 1	Slope	*a* _1_	0.02	*a* _3_	0.11
	Curvature	*a* _2_	0.03	*a* _4_	0.25 **

* *p* < 0.05, ** *p* < 0.01, *** *p* < 0.001.

## Data Availability

The data presented in this study are available from the corresponding author upon request.

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
