# Peer review of "Giving It a Shot with a Different Approach: Prosocial Strategies Moderate the Joint Effects of Agentic and Communal Goals on Bullying"

_behavsci, 2024, doi:10.3390/bs14070583_

Round 1
Reviewer 1 Report
Comments and Suggestions for Authors
Dear authors,
I was happy to review your paper "Giving It a Shot with a Different Approach: Prosocial Strategies Moderate the Joint Effects of Agentic and Communal Goals on Bullying"
The topic is highly relevant and the angle of a joint effect of agency and communion, as well as the findings that support a non-pathological perspective on bullying are interesting and important.
I have, however, few questions/remarks:
1. My main issue with the current paper is the explanation of the results. The results themselves are a bit complex but very rich, however their explanation (what does it mean, how it relate to existing findings) is just scarcely discussed. For instance, the finding that "For adolescents who strongly pursue both agentic and communal goals, their bullying does not increase but instead decreases." - is in line with plenty of literature on androgynous individuals (having high scores on both agency and communion) being best adjusted and most psychologically functional. This is not explained in the discussion but rather only briefly the findings of Mayeux and Kraft are being mentioned. On a same note, the finding that there is no significant difference between those high on both and those who are indifferent to both goals is interesting and should be elaborated better.
Similarly, unmittigated communion is being just briefly mentioned once (very unclearly for an average reader who is not in this research field), whereas, in my opinion, all results should be explained through this prism. Moreover, authors should acknowledge in the discussion the differentiation between different facets of agency and communion and how they can be reflected in social goals.
2. Another thing that is not clear enough is why prosocial strategies are a moderator and coercive strategoes are not (only controlled for)?
3. Why situation with increase in communion-decrease in agency, and decrease in both is not included in the hypotheses?
4. Was Social Goals Questionnaire adapted for adolescents?
5. The popularity subdimension was excluded from analysis. However, hypothesis H2a seem to be based on correlation with popularity (page 3) and the results are also discussed in light of popularity. Please elaborate.
Minor:
Line 130, a dot is missing: "networks Coercive strategies"
Line 199 - a sentence starts with number "10 parents" should be "Ten parents"
Line 285 - The results indicated that agentic goals were positively correlated with bullying, whereas communal goals and prosocial strategies were unrelated to bullying. --> not bullying but agency should be the last word.
Reviewer 2 Report
Comments and Suggestions for Authors
The contribution of findings is clear and demonstrates considerable application to the population under study. The research has the potential to pave the way for less deterministic approaches to bullying and victimisation. The discussion of findings is coherent and reflects outcomes of the analysis. I support this paper as 'accept with minor proofing' as outlined below.
Introduction:
Line 30-39: Relevance of the movie is unclear – yes, there are links to elements of the research, but why this ‘movie’? Nevertheless, a fascinating research area, with well-defined predictions. There are formatting issues throughout that require attention prior to being accepted. I have provided some examples below, but encourage authors to consider and proof the entire article, e.g.,
· Line 60: such as being a vegetation caretaker or task supervisor
· Line 76-78: Agentic goals emphasize influence, status, and achievement and lead individuals to compete and actively pursue their own interests in social interactions.
· Line 100-101: other[20–22]. The extant literature indicates that the correlation between the two social goals is a matter of contention[7,15,19,23–27].
· Line 129-132: making suggestions, offering assistance, and building social relationships and networks Coercive strategies entail the utilization of coercive means to directly obtain resources, such as the forcible appropriation of toys or the issuing of threats, and the exertion of control over resources through direct coercive actions.
Repeating predictions is unnecessarily repetitive. Predictions work well within paragraph corresponding to the literature as opposed to being in included in the ‘present study’.
Participants and procedure:
What do the authors mean by valid participants? Was there an inclusion criteria? This needs clarity.
The remainder of the ‘methods’ section requires proofing.
All remaining section require proofing.
Reviewer 3 Report
Comments and Suggestions for Authors
Reviewer 4 Report
Comments and Suggestions for Authors
Dear Authors,
You have done an excellent job, your involvement and motivation in the development of the work is evident. I consider that the work is quite complete and that there are few aspects that need to be improved. Nevertheless, perhaps to facilitate the reading and understanding of the readers it could be significant that the hypotheses were exposed simpler. For example, they could put the hypothesis in a simpler way and then develop it more extensively as they do.
Once you improve this detail I believe the manuscript would be ready for publication.
I wish you success in your work,
Best regards.
Round 2
Reviewer 1 Report
Comments and Suggestions for Authors
Thank you for incorporating all suggestions. Manuscript is improved with other reviewer's comments as well.
Author Response
Thank you so much for your suggestions! It dramatically enhances the quality of the paper.
Reviewer 3 Report
Comments and Suggestions for Authors
The authors have addressed my previous comments in a sufficient manner. I have only one small remark: in the hypotheses, the auhtors still write 'increases in bullying', but this should be 'higher levels of bullying'.
Comments on the Quality of English Language
Some minor editing might be needed.
Author Response
Reviewer 3
The authors have addressed my previous comments in a sufficient manner. I have only one small remark: in the hypotheses, the authors still write ‘increases in bullying’, but this should be ‘higher levels of bullying’.
Response
Thanks for your valuable comments! Now, all the hypotheses in the text no longer use “increase” or “decrease” but instead use “higher level,” “heighten,” or others.
Some minor editing might be needed.
Thanks for your valuable comments! We proofread the entire text twice using Grammarly Premium and improved most of the content (if needed).